# Polarization of HIV-1- and CMV-Specific IL-17-Producing T Cells among People with HIV under Antiretroviral Therapy with Cannabis and/or Cocaine Usage

**DOI:** 10.3390/ph17040465

**Published:** 2024-04-06

**Authors:** Fernanda de Oliveira Feitosa de Castro, Adriana Oliveira Guilarde, Luiz Carlos Silva Souza, Regyane Ferreira Guimarães, Ana Joaquina Cohen Serique Pereira, Pedro Roosevelt Torres Romão, Irmtraut Araci Hoffmann Pfrimer, Simone Gonçalves Fonseca

**Affiliations:** 1Instituto de Patologia Tropical e Saúde Pública, Universidade Federal de Goiás, Goiânia 74605-050, GO, Brazil; fofeitosa1@yahoo.com.br (F.d.O.F.d.C.); adrianaguilarde@gmail.com (A.O.G.); luizcarlosssbr@hotmail.com (L.C.S.S.); 2Escola de Ciências Médicas e da Vida, Pontifícia Universidade Católica de Goiás (PUC-Goiás), Goiânia 74605-140, GO, Brazil; 3Hospital das Clínicas Dr. Serafim de Carvalho, Jataí 75805-020, GO, Brazil; regyanefg@hotmail.com; 4Hospital de Doenças Tropicais, Goiânia 74850-400, GO, Brazil; anajoaquinapereira@uol.com.br; 5Laboratório de Imunologia Celular e Molecular, Programa de Pós-Graduação em Ciências da Saúde, Programa de Pós-Graduação em Biociências, Universidade Federal de Ciências da Saúde de Porto Alegre, Porto Alegre 90050-170, RS, Brazil; pedror@ufcspa.edu.br; 6iii-INCT-Instituto de Investigação em Imunologia, Instituto Nacional de Ciência e Tecnologia, São Paulo 05403-900, SP, Brazil

**Keywords:** HIV-1, drug addiction, cannabis, cocaine, IL-17A, IFN-γ, Th17/Tc17 cells

## Abstract

Objective: This study evaluated the influence of cannabis and/or cocaine use in human immunodeficiency virus (HIV)- and cytomegalovirus (CMV)-specific T-cell responses of people with HIV (PWH). Results: There was a higher percentage of IL-17-producing HIV-Gag-specific CD8+ T-cells in all drug users than that in PWH non-drug users. Stratifying the drug-user groups, increased percentages of IL-17-producing HIV-Gag-specific CD4+ and CD8+ T-cells were found in PWH cannabis plus cocaine users compared to PWH non-drug users. In response to CMV, there were higher percentage of IL-17-producing CMV-specific CD8+ T-cell in PWH cocaine users than that in PWH non-drug users. Considering all drug users together, there was a higher percentage of SEB-stimulated IL-17-producing CD4+ T-cells than that in PWH non-drug users, whereas cannabis users had higher percentages of IL-17-producing CD4+ T-cells compared to non-drug users. Methods: Cryopreserved peripheral blood mononuclear cells from 37 PWH undergoing antiretroviral therapy (ART) using cannabis (10), cocaine (7), or cannabis plus cocaine (10) and non-drug users (10) were stimulated with HIV-1 Gag or CMV-pp65 peptide pools, or staphylococcal enterotoxin B (SEB) and evaluated for IFN-γ- and/or IL-17A-producing CD4+ and CD8+ T-cells using flow cytometry. Conclusions: Cannabis plus cocaine use increased HIV-specific IL-17 producing T-cells and cocaine use increased IL-17 CMV-specific CD8+ T-cell responses which could favor the inflammatory conditions associated with IL-17 overproduction.

## 1. Introduction

There is a growing body of evidence suggesting that illicit drug use has systemic consequences on the body and affects disease progression in human immunodeficiency virus 1 (HIV-1) infection [1,2]. Cocaine use can modulate HIV-1 replication in monocytes, dendritic cells, and quiescent T cells [3,4,5], thus increasing the risk of progression to acquired immune deficiency syndrome (AIDS) [6]. A prospective longitudinal study with people with HIV (PWH) drug users concluded that crack-cocaine facilitates HIV disease progression with a decrease in CD4+ T-cell counts and an increase in viral load [6] independently of antiretroviral treatment (ART), age, gender, and years since HIV diagnosis. Moreover, it has been observed that crack-cocaine plus cannabis use was a multidrug combination that significantly increased the risk of HIV disease progression with a decrease in CD4+ T-cell counts [7,8]. In vitro evaluation of the cellular and molecular action of cocaine after HIV infection has shown that the administration of cocaine along with HIV infection led to an increased rate of T-cell apoptosis [9]. In relation to cannabis, heavy marijuana use has been shown as a risk factor for the incidence of cardiovascular events, such as myocardial infarction, ischemic heart disease, and heart disease in men living with HIV, independent of tobacco smoking and traditional risk factors [10]. However, cannabis usage did not show any effect on CD4+ T-cell counts in HIV and hepatitis C virus coinfected individuals [11]. An in vitro study showed that delta-9-tetrahydrocannabinol (THC), a stimulant compound from *Cannabis sativa*, decreased lymphocyte activation as well as Th1/Th17 lineage commitment in an animal model of delayed-type hypersensitivity [12].

The protection from infection can be determined by the functional persistence of antigen-specific CD4+ and CD8+ effector and memory T cells that play an important role in protection against HIV and opportunistic cytomegalovirus (CMV), *Mycobacterium tuberculosis*, *Candida albicans*, and other pathogens [13,14,15]. However, in HIV infection, excessive immune activation poses a constant challenge for CD4+ and CD8+ memory T cells, leading to immune exhaustion and the loss of the ability to respond to HIV antigens, evidenced by the expression of markers such as cytotoxic T-lymphocyte-associated protein 4 (CTLA-4) and programmed cell death protein 1 (PD-1), and immune dysfunction [16,17,18].

IFN-γ-producing CD4+ T cells (Th1) secrete ample amounts of IFN-γ, which activates a range of interferon-stimulated genes (ISG) with antiviral functions [19,20,21]. Another important T-cell subset is the IL-17-producing CD4+ T cells (Th17), which is important for protecting against extracellular pathogens, maintaining the intestinal epithelial barrier integrity through IL-17 production, and promoting the proliferation of enterocytes [22,23]. IL-17-producing CD8+ T cells (Tc17) also show potential to respond to an inflammatory environment and maintain intestinal integrity [23,24,25]. However, the accumulation and hyperactivation of Th17 and Tc17 cells can be detrimental to the homeostasis of the immune responses because both subsets are known to be central effector cells involved in the pathogenicity of multiple sclerosis, psoriasis, rheumatoid arthritis, and ulcerative colitis [26,27,28,29].

The level of immune activation during HIV infection is the strongest predictor of disease progression to AIDS [30]. Immune activation is characterized by high levels of circulating pro-inflammatory cytokines and chemokines with increased expression of CD38 on CD4+ T cells [31,32], and impaired T-cell function [33]. HIV-specific CD4+ and CD8+ T-cell exhaustion includes the progressive inhibition of cytokine production, proliferation capacity, and cytotoxicity, which leads to defective pathogen clearance in chronic viral infections, thus contributing to immunopathogenesis [34,35].

It is unclear whether the use of illicit drugs affects the functional diversity of CD4^+^ and CD8^+^ T cells in HIV infection. The present study aimed to evaluate whether the use of cannabis and/or cocaine interferes with the pool of specific T-cell responses in PWH, by assessing the functional capacities and differentiation profiles of CD4+ and CD8+ T cells after HIV, CMV, and staphylococcal enterotoxin B (SEB) stimulation in vitro, and IFN-γ- and/or IL-17-producing cells were evaluated.

## 2. Results

### 2.1. Clinical and Demographic Characteristics of Participants

A total of 37 people with HIV undergoing ART were included in the present study. The different groups under investigation were 10 PWH non-drug users (control group), 10 cannabis, seven cocaine, and ten cannabis plus cocaine users. The route of cannabis use was smoked, and cocaine was inhaled or smoked (crack).

The demographic and clinical characteristics of subjects are summarized in Table 1. The majority of PWH were male (86.48%). All individual drug users and non-drug users had undetectable viral loads (<40 copies/mL) and there were no significant differences in the age, CD4+ T-cell counts and nadir CD4+ T-cell counts among the groups (Table 1).

### 2.2. Increased IL-17-Producing HIV-Specific T-Cell Responses in PWH Drug Users

The present study aimed to determine the influence of drug use on HIV-specific T-cell responses in PWH. To investigate that, peripheral blood mononuclear cells (PBMC) were stimulated with HIV Gag peptide pool and evaluated for IFN-γ- and/or IL-17A-producing CD4+ and CD8+ T-cells using flow cytometry. The gating strategy analysis used for flow cytometry is presented in Figure 1. The analysis of CD4+ T-cell specific immune responses is shown in Figure 2 and the specific immune response of CD8+ T cells is represented in Figure 3. There were no differences between the percentages of IFN-γ-producing CD4+ (Figure 2A) and CD8+ T (Figure 3A) cells in response to HIV Gag peptide pool stimulation in all drug users taken together or in stratified drug-user groups compared to PWH non-drug users. Cells from PWH cannabis plus cocaine users under HIV Gag peptide pool stimulation showed higher percentages of IL-17-producing CD4+ (*p* = 0.029) and CD8+ T cells (*p* = 0.029) compared to PWH non-drug users (Figure 2A and Figure 3A, respectively). Moreover, an increased percentage of IL-17-producing CD8+ T cells (*p* = 0.027) was observed in response to HIV Gag peptide pool stimulation for all drug users taken together compared to PWH non-drug users (Figure 3A). There were no differences in the percentages of double cytokine (IFN-γ and IL-17)-producing CD4+ (Figure 2A) and CD8+ T cells (Figure 3A) in response to HIV Gag peptide pool among PWH drug users and non-drug users.

### 2.3. Increased CMV-Specific CD8+ T-Cell Response in PWH Cocaine Users

Considering that all individuals showed IgG anti-CMV positive results, we investigated the specific immune response to CMV, which is a common infection in PWH. CD4+ and CD8+ T-cell responses were evaluated after CMVpp65 peptide pool stimulation based on IFN-γ and IL-17 production. The percentages of IFN-γ-producing CD4+ and CD8+ T cells in response to CMVpp65 peptide pool stimulation were similar when all drug users were considered together or in stratified drug-user groups (Figure 2B and Figure 3B, respectively) compared to PWH non-drug users. The percentages of IL-17-producing CD4+ T cells in response to CMVpp65 were similar among PWH drug users and non-drug users (Figure 2B). However, PWH cocaine users showed a higher IL-17-producing CD8+ T-cell response (*p* = 0.042) than that of PWH non-drug users (Figure 3B). There were no differences in the percentages of double cytokine (IFN-γ and IL-17)-producing CD4+ (Figure 2B) and CD8+ T-cells (Figure 3B) in response to CMVpp65 among PWH drug users and non-drug users.

### 2.4. T-Cell Responses from HIV-Infected Drug Users after SEB Stimulation

CD4+ and CD8+ T-cell responses from PWH cannabis plus cocaine drug users after polyclonal stimulation with SEB showed IFN-γ production similar to that detected in cells from PWH non-drug users (Figure 2C and Figure 3C, respectively). Considering all drug users together, a higher percentage was found in SEB-stimulated IL-17-producing CD4+ T cells (*p* = 0.037) than that of PWH non-drug users (Figure 2C). Stratifying the drug-user groups, cannabis users showed higher percentages of IL-17-producing CD4+ T-cells than PWH non-drug users (*p* = 0.014) (Figure 2C). Considering PWH drug users together and stratified, similar percentages of SEB-stimulated IL-17-producing CD8+ T-cells were found compared to PWH non-drug users (Figure 3C). There were no differences in the percentages of double cytokine (IFN-γ and IL-17)-producing CD4+ and CD8+ T-cells among PWH drug-user and non-drug users after SEB stimulation (Figure 2C and Figure 3C).

### 2.5. Frequencies of Antigen-Specific Cytokine-Producing CD4+ and CD8+ T Cells in PWH Drug Users

From the total population of cytokine-producing CD4+ and CD8+ T-cells, the frequencies of IFN-γ, IL-17, and IFN-γ+IL-17+ producing CD4+ and CD8+ T cells under HIV peptide pool, CMV peptide pool, and SEB stimulation for each individual of each group were evaluated and represented in Figure 4 and Figure 5, respectively. The CD4+ T cells from the group of PWH users of cocaine showed a higher frequency of IL-17 producing cells in response to HIV peptide pool stimulation compared to frequencies of IFN-γ cells and IFN-γ+IL-17+ cells (*p* = 0.0012 and *p* = 0.0002, respectively) (Figure 4A). No differences were found in the frequencies of cytokine-producing CD4+ T cells in response to CMV peptide pool stimulation in all groups (Figure 4B). When cells were stimulated with SEB, higher frequencies of IFN-γ producing cells were observed in PWH non-drug users compared to the frequencies of IL-17-producing cells (*p* < 0.0001) and IFN-γ+IL-17+ cells (*p* < 0.0001). In the cannabis group, higher frequencies of IFN-γ producing cells and IL-17-producing cells compared to the frequencies of IFN-γ+IL-17-producing cells (*p* = 0.0017 and *p* = 0.023, respectively) were observed. In the cannabis plus cocaine user group, higher frequencies of IFN-γ-producing cells compared to IL-17- and IFN-γ+IL-17-producing cells (*p* = 0.0141 and *p* = 0.001, respectively) were detected (Figure 4C).

In relation to the compartment of CD8+ T cells, stimulation with the HIV Gag peptide pool induced a higher frequency of IL-17-producing CD8+ T cells compared to the IFN-γ (*p* = 0.0010) and IFN-γ+IL-17+ (*p* = 0.0008) frequencies in the PWH cannabis plus cocaine group (Figure 5A). After CMVpp65 pool stimulation, there was a predominance of IL-17-producing CD8+ T cells in PWH cannabis drug users compared to IFN-γ+IL-17+ cells (*p* = 0.0389) (Figure 5B). Furthermore, following CMV peptide stimulation, higher frequencies of IL-17 producing CD8+ T cells compared to the frequencies of IFN-γ producing cells (*p* = 0.002) and IFN-γ+IL-17 producing cells (*p* < 0.0001) were observed in the group of PWH cocaine users (Figure 5B). The frequency of SEB-stimulated cytokine-producing CD8+ T cells was similar among all groups, showing higher frequencies of IFN-γ producing cells compared to IL-17-producing cells and IFN-γ+IL-17 producing cells (*p* < 0.0001) in all groups, including PWH non-drug users (Figure 5C).

## 3. Discussion

The present study investigated the effects of cannabis, cocaine, and cannabis plus cocaine use on HIV- and CMV-specific CD4+ and CD8+ T-cell responses based on IFN-γ and/or IL-17 production in PWH. No changes in the percentages of IFN-γ-producing HIV- and CMV-antigen-specific CD4+ and CD8+ T-cells were observed in all drug users together or in stratified groups of PWH drug users compared to non-drug users. The SEB stimulation, a control of polyclonal cell stimuli response, showed a higher production of IFN-γ-producing cells in all drug-user groups, indicating that these cells are functionally capable of producing this cytokine, even though the occurrence did not reach statistically significant differences between the drug-user groups and the non-drug user group. Therefore, the activation pathway for IFN-γ production was not compromised in these drug users.

All drug users showed increased percentages in IL-17-producing HIV-specific CD8+ T cells compared to the PWH non-drug users, but the same was not observed in the HIV-specific CD4+ T-cell response in all drug users. The evaluation of drug-user groups separately indicated a polarization to Th17 (CD4+) and Tc17 (CD8+) HIV-specific responses in the cannabis plus cocaine PWH users compared to the PWH non-drug user group. IL-17 is an important cytokine involved in protecting against extracellular pathogens in mucosal barrier surfaces and in the regulation of intestinal epithelial cell permeability [36]. Moreover, a subset of Th17 cells from blood expressing CCR4+CCR6+ was permissive to HIV replication in vitro. These cells harbored high levels of HIV DNA and were long living [37]. In our study, we observed an increase in IL-17 producing cells in cannabis plus cocaine PWH users. These cells are targets of HIV infection. In addition to the increase in inflammation observed in our previous work [38], it is possible to suggest that these PWH could be susceptible to aggravation of HIV disease and to developing inflammatory disorders. High percentages of IL-17-producing T-cells and IL-17-promoting cytokines are associated with persistent heart failure and are also implicated in multiple sclerosis and psoriatic lesions, contributing to disease severity [39,40]. Indeed, we observed higher levels of high-sensitivity C-reactive protein, a biomarker of heart failure [41], in the plasma of cannabis plus cocaine PWH users in our previous work [38], reinforcing the possibility of disease aggravation in those individuals. In several pathologies, particularly autoimmune diseases, pathogenic and non-pathogenic subsets of Th17 cells have been identified [42,43,44,45,46], characterized by the expression of chemokine/cytokine receptors, cytokine production, and other molecules. Furthermore, studies have shown that the cytokine IL-23 plays a crucial role in driving Th17 cell plasticity toward a more pro-inflammatory profile, thereby serving as an essential contributor to Th17 cell-mediated pathogenicity [42,43,46]. Th17 and Tc17 cells have features that might contribute to immunoactivation and the maintenance of the HIV reservoir, suggesting a pathogenic role for these cell subsets [47,48]. However, we did not investigate the size of HIV reservoir in Th17 cells among PWH drug users, nor did we further characterize the phenotype of pathogenic and non-pathogenic Th17 to better elucidate these points. We can suggest a potential involvement of pathogenic Th17 cells, based on the observed inflammatory environment in PWH who use drugs, compared to those who do not [38].

In this study, we also observed an increase in IL-17-producing CD4+ T-cells in the drug-user groups under SEB stimulation. This augmentation could be related to the increased percentages of IL-17-producing CD4+ T-cells observed in the cannabis user group compared to PWH non-drug users under the same stimulus. However, we cannot explain this cannabis reaction, since it has been reported that the cannabis has an anti-inflammatory pattern [49,50]. Some points that may be relevant to an explanation of these discrepancies are the quantity and intensity of use of the cannabis by individuals. Other studies have evaluated the effect of cannabis on inflammation in other types of inflammatory diseases [51].

In our study, cells stimulated with a CMV peptide pool showed an increase in the percentages of IL-17-producing CMV-specific CD8+ T-cells in the group of PWH cocaine users compared to PWH non-drug users. The percentages of IFN-γ-producing specific CD8+ T cells in the same group showed no differences compared to PWH non-drug users. The deviation to augment of IL-17-producing CD8+ T-cells in the cocaine user group in relation to IFN-γ-producing CD8+ T cells could contribute to impairment of the control of the CMV infection in PWH. It has been shown that the high cytotoxic effect of CMV-specific CD8+ T-cells, which are producers of IFN-γ, is maintained during the latent phase of CMV infection to prevent virus reactivation and protect against re-infection [52]. CMV infection in immune-compromised individuals may cause severe non-AIDS-defining events [53]. Moreover, PWH who are unable to control CMV replication have an increased risk of HIV disease progression [54]. Based on the data reported above and in addition to our results, we can hypothesize that the augmentation of IL-17-producing CD8+ T-cells induced by cocaine use may facilitate the CMV reactivation in PWH. In this context, it will be difficult to control a CMV infection that is mediated by the IFN-γ producing CD8+ T-cells. This result indicates that the use of cocaine could result in a pro-inflammatory outcome [55,56]

In our study, we did not find changes in the percentages of double-cytokine-producing CD4+ and CD8+ T-cells among the groups under HIV, CMV, or SEB stimulation. Th1/Th17 cells that exhibit either T-box transcription factor (Tbet) and retinoic acid-related orphan receptor-gamma t transcription factor (RORγt) are able to produce both IFN-γ and IL-17A [57]. Thus, when they are exposed to specific polarization stimuli, they have the capacity to acquire either Th1 or Th17 profiles [58]. Cytokines are the main factors that determine the lineage of T helper cell differentiation. A recent work from our group showed a pro-inflammatory profile in relation to the IL-6/IL-10 plasma level ratio in PWH cannabis plus cocaine users compared to PWH non-drug users [38]. IL-6 is one of the key cytokines for the induction of Th17 differentiation. Data from the present study suggest that the use of cannabis plus cocaine by PWH could favor an inflammatory environment that might induce the activation of RORγt, leading to Th17 and Tc17 differentiation as observed by others [58].

The frequency analysis of cytokine-producing cells in response to HIV peptide pool stimulation in each group of PWH drug users showed higher frequencies of IL-17 producing CD4+ T cells in the cocaine group and IL-17 producing CD8+ T cells in the cocaine plus cannabis group, in comparison to other cytokine-producing cells. In relation to the CMV-specific response, the frequencies of IL-17 producing CD8+ cells were higher in the cannabis and cocaine groups compared to the frequencies of IFN-γ and IFN-γ+IL-17 producing cells. However, in both the non-drug user and drug user groups among PWH, polyclonal stimulation resulted in a predominant frequency of IFN-γ producing cells. This suggests that the IL-17 response is associated with an antigen-specific T-cell response in PWH drug users. The reasons for the increase of IL-17 antigen-specific producing cells in drug users is not well established. A recent in vitro study found that diesel exhaust and cigarette smoke extracts in culture experiments enhanced Th17 responses through the aryl hydrocarbon receptor (AhR) [59], which has been shown to drive the differentiation of Th17 cells [60]. Considering that cannabis and cocaine are usually smoked, this may imply that the components present in the smoke produced by those drugs could contribute to the immunodeviation to IL-17 producing cells. Moreover, a recent study showed that AhR can constitute a barrier to HIV-1 replication in CD4^+^ T cells following the activation of a T-cell receptor in vitro [61]. Further studies are necessary to elucidate the role of AhR in the context of drug abuse.

Our study has several strengths, including the stratification of PWH who are drug users, categorized as users of cannabis, cocaine, or a combination of cannabis and cocaine, using a urine drug screening test. This stratification made it possible to evaluate the effects of either a single drug or a combination of drugs on antigen-specific T-cell responses. Another important point is that all PWH were on ART with undetectable viral loads, which allowed us to avoid the effects of HIV replication. The fact that all PWH were positive for CMV led to the investigation of the effects of drug use on a pathogen other than HIV, enriching our work. However, our study does have several limitations, such as the low number of participants in each group and the absence of information regarding the quantity of drug use and alcohol consumption for each individual. Additionally, adjusted multiple comparisons for paired comparisons for all outcomes were not conducted. Nevertheless, this study is of great importance because it demonstrates that the use of illicit drugs has an impact on the specific immune response in PWH under antiviral treatment, leading to a shift in the antiviral response from the Th1 type to the Th17 type.

## 4. Material and Methods

### 4.1. Study Participants and Sample Collection

Our study included PWH who were non-users or users of cannabis and/or cocaine, under ART from HIV outpatient clinics of the Clinic Hospital of the Federal University of Goiás (HC/UFG), the Tropical Diseases Hospital (HDT/GO), and the Jataí Municipal Medical Center, Goiás, Brazil. We employed convenience sampling to recruit study participants. Individuals were invited to participate in the study during their routine follow-up doctor visits if they were people with HIV (PWH), at least 18 years old, and receiving antiretroviral therapy (ART) with an undetectable viral load. For PWH who were also drug users, we invited individuals meeting the above-mentioned criteria and with a history of illicit drug use as identified in medical records or indicated by their doctors. Participants were also tested to determine CMV serology. Further screening involved the examination of medical records and undertaking interviews. The PWH drug users were chronic users, but did not report intravenous drug use. Drug users were defined by self-declaration and confirmation by positive toxicological testing from urine samples indicating acute use of drugs. The exclusion criteria included vaccination in the 30 days prior to the study, acute infection at the time of sample collection, and not undergoing ART. A total of 37 PWH successfully treated subjects were included and divided into four groups: 10 non-drug users, ten cannabis, 7 cocaine, and ten cannabis plus cocaine users. Blood samples were collected in tubes containing heparin (Becton & Dickinson, San Diego, CA, USA) to prepare PBMCs and urine samples were collected in sterile flasks for rapid toxicological testing.

### 4.2. Toxicological Testing

Urine samples were used to perform a rapid toxicological immunoassay (One Step Multi-Drugs, Multi-Line Dispositive Screen Test—ABON^R^ Multidrug, Abbott, Chicago, IL, USA) for the qualitative detection of twelve drugs. Patients only positive for cannabis (THC) and/or cocaine, with the following respective cut-off points, 50 ng/mL and 300 ng/mL, were included in the study. PWH non-drug users were negative for all drugs.

### 4.3. Immunoenzymatic Assay for Anti-CMV IgG

Plasma samples were screened for the IgG antibody to CMV using a commercial immunoassay kit (CMV IgG; DIA.PRO Diagnostic Bioprobes SRL, Milan, Italy) according to the manufacturer’s instructions. The results were recorded as international units per milliliter (IU/mL). Values greater than 0.5 IU/mL were considered positive for IgG anti-CMV.

### 4.4. Cell Preparation

PBMCs from PWH non-drug users and cannabis and/or cocaine users were isolated using Ficoll-Hypaque gradient centrifugation (density 1.077 g/L, Ficoll-Paque™Plus, GE Healthcare-Uppsala-Sweden). Cell concentration and viability (>98% viability) were determined using trypan blue staining (Gibco, Grand Island, NE, USA). Cells were resuspended in a solution of fetal bovine serum (FBS) (Sigma-Aldrich, St. Louis, MO, USA) with 10% of dimethyl sulfoxide (DMSO, Sigma-Aldrich, St. Louis, MO, USA) and cryopreserved in liquid nitrogen.

### 4.5. PBMC Stimulation with HIV-1, CMV Peptide Pools, or SEB

Cryopreserved PBMCs were thawed in RPMI-1640 (Lonza, Walkersville, MD, USA) supplemented with 10% FBS (Sigma-Aldrich, St. Louis, MO, USA) and 0.1% penicillin/streptomycin (Gibco, Grand Island, NY, USA), and rested at 37 °C with 5% CO_2_ for 3 h. PBMCs (2 × 10^6^ cells) were cultured in polypropylene tubes (5 mL, Falcon^®^ Corning, Corning, NY, USA) in fresh complete medium at 37 °C with 5% CO_2_ for 5 h, with or without stimulation with previous standardized concentrations of a peptide pool of HIV-1 consensus B Gag including 123 peptides of 15mers with 11 amino acid overlaps (18 µg/mL; NIH AIDS Reagent Program: 12425), as used previously [62]; a pool of human CMV peptides comprising 138 peptides covering the entire pp65 protein of HCMV (most of the peptides are 15 amino acids lengths with 11 amino acid overlaps between sequential peptides) (70 µg/mL; HCMVpp65 peptide set, NIH AIDS Reagent Program: 12014) or Staphylococcal enterotoxin B from *Staphylococcus aureus* (SEB) (2 µg/mL; Sigma-Aldrich). Brefeldin A (1 µg/mL, eBioscience, San Diego, CA, USA) was added 1 h after stimulation and cells were further cultured for 4 h. Non-stimulated control cells were incubated with DMSO at the same concentration as that used to dilute the peptide pools. Cellular phenotypes and cytokine production were evaluated by flow cytometry.

### 4.6. Cell Surface and Intracellular Cytokine Staining by Flow Cytometry

After stimulation, PBMCs from all subjects included in the study were stained for cell surface markers with fluorochrome-conjugated antibodies against cell surface CD3 (V450, clone: UCHT-1, cat#580365, BD Pharmingen, San Diego, CA, USA), CD4 (PercP-Cy5.5, clone:S3.5, cat#MHC00422, Caltag/Thermo Fisher Scientific, Waltham, MA, USA), and CD8 (Pacific blue, clone:RPA-T8, cat#558207, BD Pharmingen, San Diego, CA, USA) and LIVE/DEAD^®^ labeling. Cells were then fixed and permeabilized with Cytofix/Cytoperm according to the manufacturer’s instructions (BD Biosciences, San Jose, CA, USA), followed by intracellular staining with fluorochrome-conjugated antibodies against IFN-γ (PE, clone: 4SB3, cat#559326, BD Pharmingen) and IL-17A (PE-Cy7, clone: BL168, cat#512315, Biolegend, San Diego, CA, USA). For all cellular phenotyping by flow cytometry, fluorescence minus one (FMO) controls were used to set gates for positive and negative staining. After staining, the samples were acquired on a BD FACSCanto™ II flow cytometer (BD Biosciences, San Diego, CA, USA) using FACSDIVA version 8 software (BD Biosciences, San Diego, CA, USA). A minimum of 400.000 events was recorded from each tube. Data were analyzed using FlowJo Software (version 10.0, BD Biosciences, San Diego, CA, USA).

### 4.7. Data Analysis

The non-parametric Mann–Whitney test was used to analyze differences between unpaired groups, comparing each PWH drug-user group to PWH non-drug users. Fisher’s exact test was used to compare a categorical variable (sex) of each PWH drug-user group to PWH non-drug users. The percentages of cells involved in the three distinct profiles of IFN-γ, IL-17 and IFN-γ+IL-17 production were calculated in each group of drug users. We calculated the responsiveness of cells stimulated with peptide pools by subtracting the results from the unstimulated condition (DMSO). To calculate the frequencies of cytokine-producing cells, the sum of the total of IFN-γ, IL-17 and IFN-γ+IL-17 positive cells was considered one hundred percent in each individual of each group of drug users. Then the frequencies of IFN-γ and/or IL-17 CD4+ and CD8+ T cells were established. The results were presented as the individual frequency of each cytokine, and the mean frequencies ± Standard Error of the Mean (SEM) were shown. For statistical analysis of frequencies, the one-way ANOVA with multiple comparisons and Tukey’s post-hoc testing was used. Statistical analyses were performed using Prism version 5.0 (GraphPad Software Inc., San Diego, CA, USA). *p* values < 0.05 were considered statistically significant.

## 5. Conclusions

In conclusion, the use of cannabis plus cocaine increased HIV-specific IL-17 producing T cells, and cocaine use increased the IL-17 CMV-specific CD8+ T-cell response. The augmentation of the frequencies of IL-17-producing T cells in PWH drug-user groups could favor the inflammatory conditions associated with IL-17 overproduction. Further studies are needed to clarify the role of IL-17-producing T cells in the context of drug abuse in HIV infection.

## Figures and Tables

**Figure 1 pharmaceuticals-17-00465-f001:**
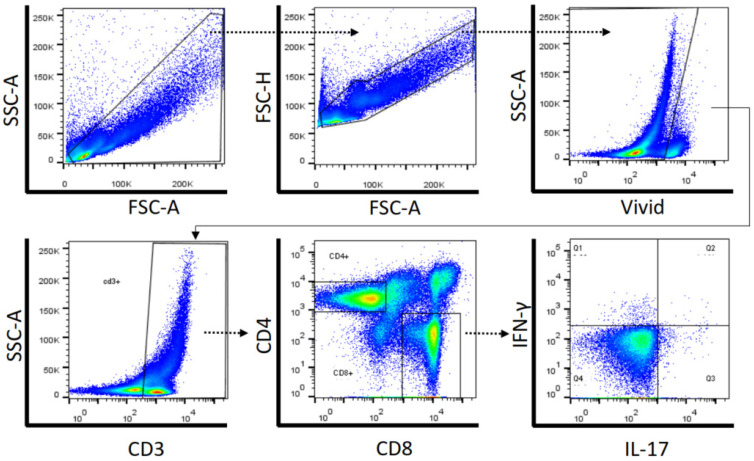
Representative gating strategy used for lymphocyte identification by flow cytometry using forward and side scatter properties. Lymphocytes were gated on CD3+ and the percentages of CD4+ and CD8+ T lymphocytes were determined. In each gate of CD4+ and CD8+ T cells, the percentages of IFN-γ and IL-17-producing T cells were determined. Dead cells were excluded using LIVE/DEAD^®^ labeling (ViViD).

**Figure 2 pharmaceuticals-17-00465-f002:**
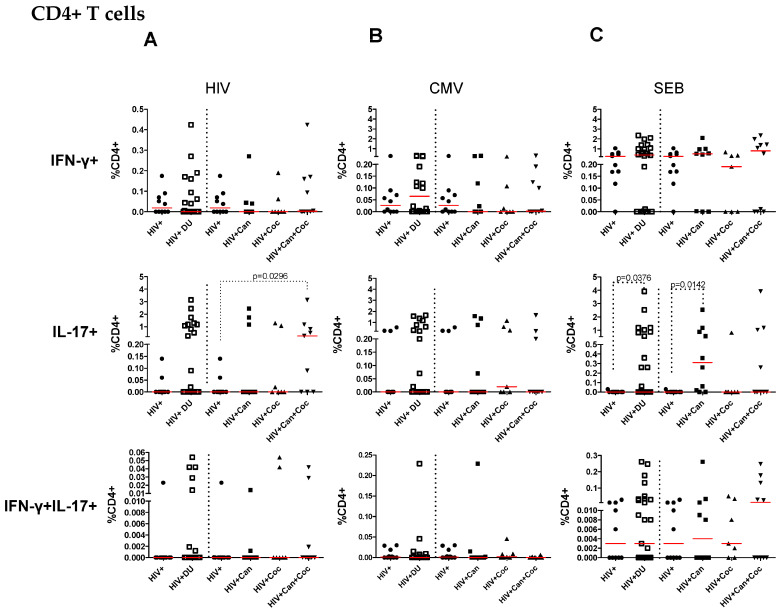
Percentages of IFN-γ-, IL-17A- and double cytokine-producing CD4+ T cells in response to in vitro HIV Gag peptide pool, HCMVpp65 peptide pool, and SEB stimulation. The percentages of IFN-γ-, IL-17-, and double cytokine-producing CD4+ T cells in PWH non-drug users (n = 10), all drug users taken together (HIV + DU), and in each group of drug users (cannabis = 10, cocaine n = 7, and cannabis plus cocaine n = 10) were determined after in vitro stimulation with HIV Gag peptide pool (**A**), HCMVpp65 peptide pool (**B**) and SEB (**C**). The Mann–Whitney test was used to compare the percentages of cells among drug-user groups and control groups. Percentages of cytokines producing cells after peptide pool stimulation were calculated after subtracting values of control condition (i.e., dimethyl sulfoxide -DMSO). *p* values < 0.05 were considered statistically significant. All *p* values are shown in Appendix A.

**Figure 3 pharmaceuticals-17-00465-f003:**
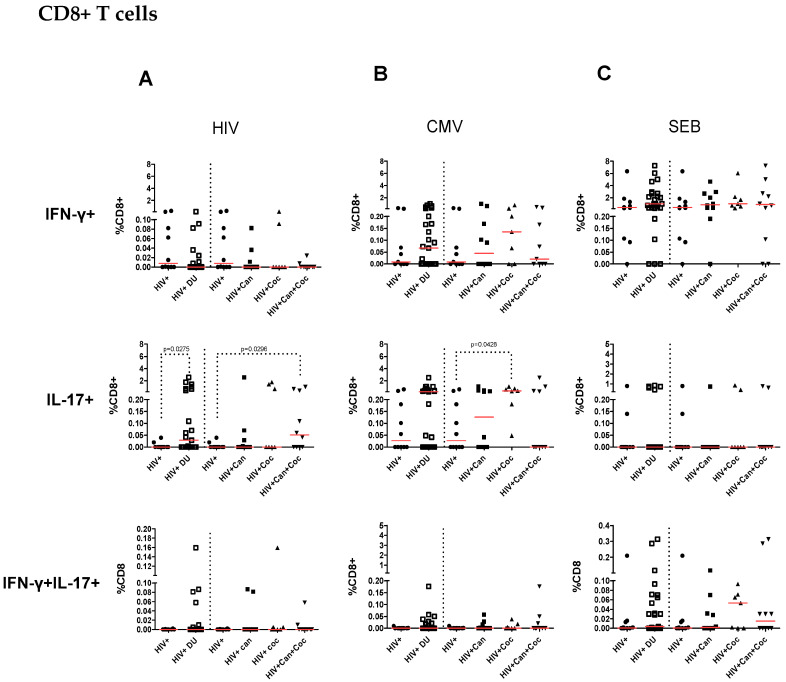
Percentages of IFN-γ-, IL-17A-, and double cytokine-producing CD8+ T cells in response to in vitro HIV Gag peptide pool, HCMVpp65 peptide, and SEB stimulation. The percentage of IFN-γ-, IL-17-, and double cytokine-producing CD8+ T (**A**) cells was determined corresponding to all drug users taken together (HIV + DU) and to each group of drug users (cannabis = 10, cocaine n = 7, and cannabis plus cocaine n = 10) and non-drug users (n = 10) after in vitro stimulation with HIV Gag peptide pool (**A**), HCMVpp65 peptide pool (**B**), and SEB (**C**). Percentages of cytokines producing cells after peptide pool stimulation were calculated after subtracting values of the control group (i.e., dimethyl sulfoxide—DMSO). The Mann–Whitney test was used to compare the percentage of cells between PWH drug users and non-drug user control group. *p* values < 0.05 were considered statistically significant. All *p* values are shown in Appendix A.

**Figure 4 pharmaceuticals-17-00465-f004:**
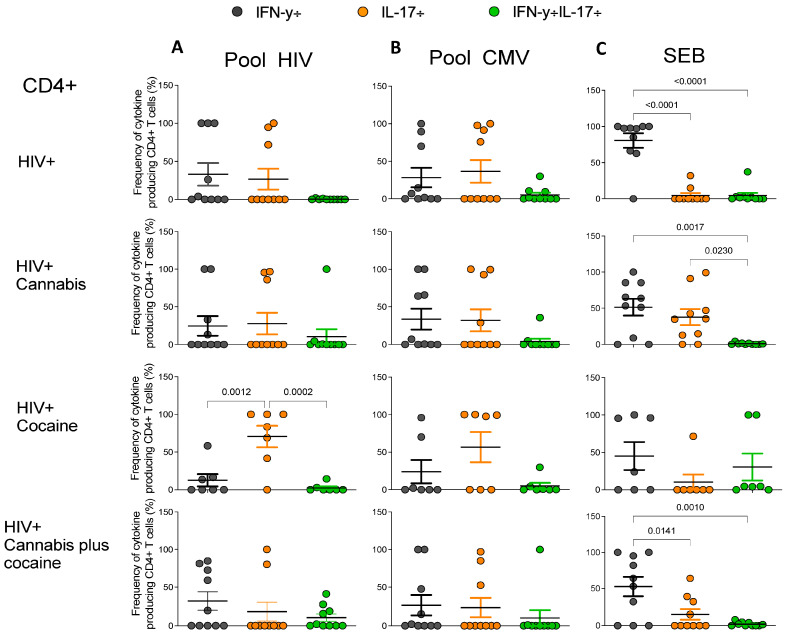
Frequencies of cytokine-producing CD4+ T cells in response to in vitro stimulation with HIV Gag peptide pool (**A**), HCMVpp65 peptide pool (**B**), or SEB (**C**) in each group are represented. Each graph depicts the individual frequency and the mean frequencies ± Standard Error of the Mean (SEM) of single and double cytokine-producing CD4+ T cells from PWH drug users (cannabis = 10, cocaine n = 7, and cannabis plus cocaine n = 10) and non-drug users (n = 10) in response to HIV Gag peptide pool, CMV, and SEB peptides based on IFN-γ, IL-17, or IFN-γ+IL-17 production. For statistical analysis, one-way ANOVA with multiple comparisons and Tukey’s post-hoc testing was used. *p* values < 0.05 were considered statistically significant.

**Figure 5 pharmaceuticals-17-00465-f005:**
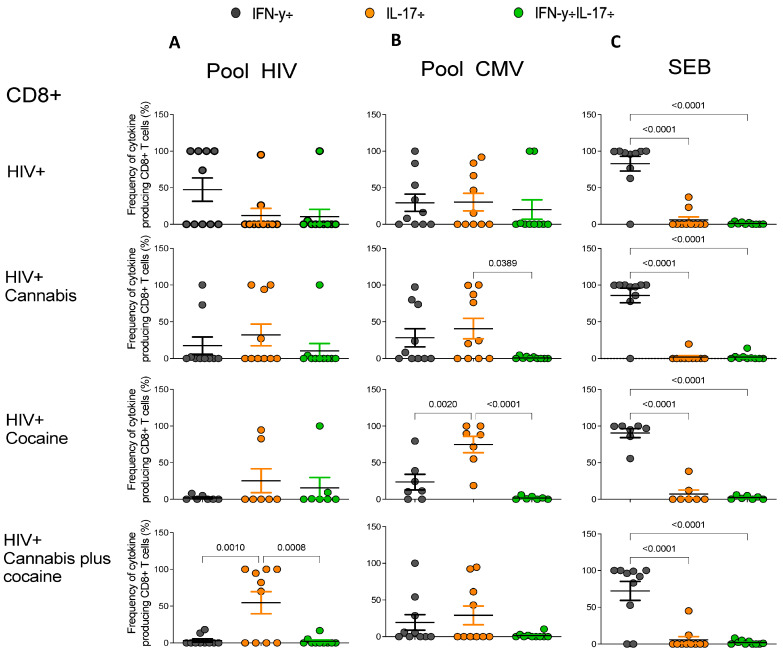
Frequencies of cytokine-producing CD8+ T cells in response to in vitro stimulation with HIV Gag peptide pool (**A**), HCMVpp65 peptide pool (**B**), or SEB (**C**) in each group are represented. Each graph depicts the individual frequency and the mean frequencies ± Standard Error of the Mean (SEM) of single and double cytokine-producing CD8+ T cells from PWH drug users (cannabis = 10, cocaine n = 7, and cannabis plus cocaine n = 10) and non-drug users (n = 10) in response to HIV Gag peptide pool, CMV, and SEB peptides based on IFN-γ, IL-17, or IFN-γ+IL-17 production. For statistical analysis, one-way ANOVA with multiple comparisons and Tukey’s post-hoc testing was used. *p* values < 0.05 were considered statistically significant.

**Table 1 pharmaceuticals-17-00465-t001:** Clinical and demographical characteristics of participants.

Characteristics	HIV Non-Drug Users (*n* = 10)	HIV Cannabis Users (*n* = 10)	HIV Cocaine Users (*n* = 7)	HIV Cannabis Plus Cocaine Users (*n* = 10)	*p* Value
Gender (male/female)	9/1	10/0	5/2	8/2	ns
Age, years Media (±SD)	39 (±9.0)	37 (±8.2)	38 (±10.9)	38 (±7.8)	ns
Smokers % (n)	30% (3)	30% (3)	42.8% (3)	40% (4)	
Years of treatment Media (±SD)	5.4 (±3.0)	4.8 (±4.3)	5.7 (±3.6)	8.7 (±6.7)	
**Antiretroviral regimen (%)**					
AZT/3TC/EFV	40.0	50.0	57.1	40.0	
AZT/3TC	30.0	-	14.3	-	
AZT/3TC/LPV/r	10.0	10.0	28.6	-	
AZT/RIT	10.0	-	-	-	
3TC/TDF/EFV	-	30.0	-	30.0	
Biovir	10.0	-	-	-	
Biovir/NVP	-	-	-	20.0	
TDF/3TC/RIT/AZT	-	10.0	-	10.0	
CD4 counts cels/mm^3^ media (±SD)	557.4 (±183)	489.1 (±270)	627 (±209)	389 (±288)	ns
HIV-1 Viral load	<LD	<LD	<LD	<LD	
Nadir CD4 (media ± SD)	343 (±255)	252 (±180)	347 (±179)	297 (±282)	ns

LD: Limit of detection. AZT: zidovudine; 3TC: lamivudine; EFV: efavirenz: RIT: ritonavir; TDF: tenofovir disoproxil fumarete; Biovir: AZT + 3TC; NVP: nevirapine; LPV/r: liponavir/ritonavir. ±SD: ±Standard Deviation; ns: not statistically significant. Categorical variable (sex) of each PWH drug-user group was compared to PWH non-drug user group using Fisher’s exact test. Continuous variables (Age, CD4 counts, nadir CD4 counts) of each PWH drug-user group was compared to PWH non-drug users group using Mann–Whitney test. *p* values < 0.05 were considered statistically significant.

## Data Availability

The data are contained within the article. Any other data can be provided by demand to the authors.

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
