# Peer review of "Polarization of HIV-1- and CMV-Specific IL-17-Producing T Cells among People with HIV under Antiretroviral Therapy with Cannabis and/or Cocaine Usage"

_pharmaceuticals, 2024, doi:10.3390/ph17040465_

Round 1

Reviewer 1 Report

Comments and Suggestions for Authors

The authors compared T cell responses (i.e., cytokines) of HIV infected subjects who were on ARV along to ARV combined with the use of cannabis along, cocaine along, both cannabis and cocaine. The authors presented the results based on statistical significance (p-values <0.05). If I am not miss-reading, the differences between groups were not presented or commented. Were the differences found based on p-value clinically\biologically meaningful or not? I am afraid that statistical significance might not be found when multiple comparisons were adjusted. Also, I suggest the authors to better align the method and results; specify the comparison groups and aligned the outcome measures with statistical method used for comparisons.

Line 115-125, what statistical method was used to compare these percentages and produce p-values? Please clarify in the method section.

Line 395-396, were Mann-Whitney test and Kruskal-Wallis tests used to compare groups for all variables in Table 1 and all figures (percentages of cytokines)?  Were these two tests used inter-changeably?

Line 396, were multiple comparisons adjusted for paired comparisons for all outcomes?  If not, please consider stated as a limitation.

Line 337-341, were these PWH recruited by the primary physicians?  When was these subjects recruited?  What were the inclusion criteria besides being at least 18 years old?  How many were recruited and how many were excluded from the study and for what reasons?

Line 400 to 402, Please clarify what the authors mean about using sum of total as 100 percent, then frequency is established. Total of what?

Author Response

Reviewer #1

We would like to thank you for your considerations that enabled us to better describe our results throughout the manuscript.

 The authors compared T cell responses (i.e., cytokines) of HIV infected subjects who were on ARV along to ARV combined with the use of cannabis along, cocaine along, both cannabis and cocaine. The authors presented the results based on statistical significance (p-values <0.05). If I am not miss-reading, the differences between groups were not presented or commented. Were the differences found based on p-value clinically\biologically meaningful or not? I am afraid that statistical significance might not be found when multiple comparisons were adjusted. Also, I suggest the authors to better align the method and results; specify the comparison groups and aligned the outcome measures with statistical method used for comparisons.

Main points noted:

  1. The authors presented the results based on statistical significance (p-values <0.05). If I am not miss-reading, the differences between groups were not presented or commented.

Answer: All the results and p-values have been presented in the text, as can be seen on pages 3, 4, and 7, 8 and 9, itens “2.2. Increased IL-17-producing HIV-specific T cell responses in PWH drug users”, “2.3. Increased CMV-specific CD8+ T cell response in PWH cocaine users”, “2.4. T cell responses from HIV-infected drug users after SEB stimulation” and “2.5. Frequencies of antigen-specific cytokine-producing CD4+ and CD8+ T cells in PWH drug users”. The comments of the results where done in discussion section (pages 9, 10, 11).  The comparison was conducted between each group of  PWH who weredrug users (cannabis, cocaine or cannabis+cocaine) and PWH  who did not use drugs. We performed one comparison at a time and did not compare the diffferent drug user groups with each other. Therefore, this description was note included in the results.

  1. Were the differences found based on p-value clinically\biologically meaningful or not?

Answer: We based our results in compare difference in response of T cells in cytokine production after antigen specific stimulation. Cells was obtained from two different defined groups of PWH, likewise, those who use illicit drugs (cannabis, cocaine or cannabis+cocaine) versus those who did not use illicit drugs. Thus, the values of percentages of producing cytokine cells from these distinct PWH group was evaluated between which other. We would like to observe if the use of illicit drugs interferes in CD4+ and CD8+ T cells function when encounter with a pathogen. We used the non-parametric Mann-Whitney test for the analyses. The differences observed in this study, based on statiscal significance, showed that the use of cannabis plus cocaine by PWH increased the percentages of HIV-specific IL-17 producing CD4+ and CD8+ T cells compared to PWH who were non-drug users. Additionally, cocaine use by PWH increased IL-17 CMV-specific CD8+T cell response compared to PWH non-drug users. The elevation in the frequencies of IL-17 cells in PWH drug-user groups could potentially contribute to inflammatory conditions associated with IL-17 overproduction, possibly leading to disease progression. Therefore, these findings are clinically and biologically significant, warranting further investigation.

  1. I am afraid that statistical significance might not be found when multiple comparisons were adjusted.

Answer: Considering that our question was the impact of the use of drug use in PWH compared to PWH non-drug user, we just did the comparison between two groups. For that, we used the non-parametric Mann-Whitney test. To answer we question about the multiple comparison, we performed the Kruskall-Wallis test, , and no statistical significance was found between the percentage of cytokine-producing cells between the study groups.

Specific points:

  1. Line 115-125, what statistical method was used to compare these percentages and produce p-values? Please clarify in the method section.

Answer: We used Mann-Whitney test to compare percentages and produce p-values. We clarified this point in the method section, like you can see on lines 494-495 page 13 and in the figure legends (line 220, page 6 and line 237, page 7).

2 - Line 395-396, were Mann-Whitney test and Kruskal-Wallis tests used to compare groups for all variables in Table 1 and all figures (percentages of cytokines)?  Were these two tests used inter-changeably?

Answer:  We included a column showing the p value of the statistical analyses in Table 1. For categoric variable (sex) we used the fisher ‘exact test and we did not find statistical significance (p>0.05). For the continuous variables (CD4 counts, nadir CD4 counts), we used the Mann-Whitney test comparing each drug user group with non-drug user group, and we did not find statistical significance (Table 1 legend), lines 109-111, page 3. For figures 3 and 4, we carried out Mann-Whitney test to compare percentages of producing cytokines cells comparing each drug user group with non-drug user group. We included this information in figures 2 and 3 legends (line 220, page 6, and line 237, page 7), and let it clear in the topic of statistical analyses in the Methods session (lines 494-497, page 13).

3 - Line 396, were multiple comparisons adjusted for paired comparisons for all outcomes?  If not, please consider stated as a limitation.

Answer: We exclusively conduct comaprisons between data from the the group of PWH non-drug users and each group of PWH drug users (cannabis, or cocaine, or cannabis plus cocaine), separately. To perform these comparisons, we employed the non-parametric Mann-Whitney test. Notably, we have not presented the data of multiple comparison as mentioned before. We took out from the text this information about Kruskal-Wallis test. Consequently, we did not adjust multiple comparisons for paired comparisons for all outcomes. This limitation is explicitly acknowledged on page 11, line 414-415.

4 - Line 337-341, were these PWH recruited by the primary physicians?  When was these subjects recruited?  What were the inclusion criteria besides being at least 18 years old?  How many were recruited and how many were excluded from the study and for what reasons?

Answer: The study subjects were recruited during their medical follow-up doctor visits. None of them were attending for the first time by the physicians. All PWH were already taking antiretroviral drugs. The inclusion criteria were the declaration of use of illicit drug form the PWH drug users group and non-use of illicit drugs for the use for the non-drug users group. Another criterion for inclusion in the study was that individual must be on antiretroviral treatment with an undetectable viral load. Additionally, a positive serology for cytomegalovirus (CMV) was required. To confirm the drug use of each participant, we used the multi-drug test (ABONTM Multidrug Test – cat no 1155990204) to detect drugs in urine. This test screens for the qualitative detection of multiple drugs and drug metabolites in human urine. PWH with a positive screening only for cannabis and/or cocaine were included in the respective groups of drug users. Non-drug use   was also confirmed using the same kit. Thus, PWH with negative screening for all drugs and metabolites detected in the toxicological test were included in the control group (PWH non-drug users). We included and clarified this information on method section (page 12, lines 431-434).

            We collected blood and urine from more PWH than were included in the experiments: 13 cannabis users, 8 cocaine, 13 cannabis plus cocaine and 16 non-drug users, a total of 13 individuals that was not included in the experiments.  The primary reason for exclusion of these individuals was the discrepancy in age and sex between the groups. Another reason was related to the detection of other drugs or metabolites apart from cannabis and cocaine, in the urine tests of PWH. We conducted this before performing the experiments. As shown in Table 1, no significant differences in variables were observed between non-drug user group and each of the drug user group.

5 - Line400 to 402, Please clarify what the authors mean about using sum of total as 100 percent, then frequency is established. Total of what?

Answer:  We aimed to observe, within each group of drug users (cannabis, cocaine and cannabis+cocaine), the predominant type of CD4+ and CD8+ T cell immune response (IFNY+, IL-17+ or IFN-Y/IL-17+) when exposed to a specific antigen (HIV-1, CMV or SEB). In other words, we sought to identify the most frequent among the three types of functional cellular responses we evaluated in each group and for each specific stimulus.  Thus, based on the total number of cytokine-producing cells in each group, that was the frequency (representation) of each type of response (IFN-Y +, IL-17 or IFN-Y /IL-17+). Total of what? The total represents the sum of all cells that produced cytokines (lL-17 + IFN-Y + IL-17/IFN- Y) following antigen stimulation. This total accounts for 100% of all cytokine-producing cells within each group. From this value, we derived the frequency of cells producing specific cytokines. We clarified this in the text, lines 501-504 page 13.

Reviewer 2 Report

Comments and Suggestions for Authors

Please see attached report.

Comments on the Quality of English Language

Please see attached report.

Author Response

Reviewer #2

 We would like to thank you for your considerations that enabled us to better describe our results throughout the manuscript.

The article by de Castro et al., is entitled “ Polarization of HIV- and CMV-specific IL-17 producing T cells in HIV-infected Cannabis plus Cocaine and Cocaine Users”. The authors examined the specific immune response of IFN-γ- and IL-17A-producing CD4+ and CD8+ T-cells in response to in vitro stimulation with HIV-1-Gag or CMV-pp65 peptide pools, or SEB. These were carried out among a total of 37 PWH under ART treatment and divided into 10 non-drug users, 10 cannabis, 7 cocaine, and 10 cannabis-plus-cocaine users. The authors found a higher frequency of IL-17-producing HIV-Gag-specific CD8+T-cells in all drug users when compared to non-drug users. Regarding CMV, there were higher IL-17- producing CMV-specific CD8+ T-cell frequencies in HIV-infected cocaine users than among non-drug users. The authors conclude that cannabis-plus-cocaine use increased HIV-specific IL-17 producing T-cells and cocaine use increased IL-17 CMV-specific CD8+ T-cell responses which could favor the inflammatory conditions associated with IL-17 overproduction. The manuscript is well-written and well-organized with minor editorial issues. However, the following concerns must be addressed by the authors before consideration for publication.  

Major concerns: 1. Abstract: The design and methods are highly redundant and can be combined into one section reducing redundancy. Also, the number of subjects in the four patient groups should be indicated. Answer: Thank you for your comment. We really agree that the information is redundant. Thus, we have removed one (design) of the repeated parts of the summary, as you can see on page 1 in the summary section. 2. Table 1: Statistical significance (p value) between the different groups should be included in the table. Also, “viral load” should be “HIV-viral load” as it can be confused with CMV viral load. Answer: Thanks for the good comment. We have made the modification as suggested and included “HIV” in viral load term in table 1. Regarding the consideration of including the significant difference (p-Value) in the table 1 between the groups, we included the results of the statistical analysis for most of the variables of Table 1 and included a column with the p value. For categoric variable (sex) we used the fisher’s exact test, and we did not find significance (p>0.05). For the continuous variables (CD4 counts, nadir CD4 counts) we used the Mann-Whitney test comparing each drug user group with non-drug user group, and we did not find statistical significance (Table 1). This information was included in the legend of Table 1 (lines 109-111 page 3). We have presented the clinical characteristics of the patients in Table 1 to demonstrate the profile of the individuals in our study. Our study groups were matched as closely as possible on the variables of age and cigarett use, for example. The other information shows the type of antiretroviral used and the undetected viral load, which were criteria for including these individuals in the study.  3. The length and overlap of HIV-1 gag and CMV pp65 peptides used in the stimulation must be indicated at least in the methods section. A supplemental table for the sequences of these pools could be helpful. Answer: The HIV-1 clade B gag peptide pool comprising 123 peptides spanning the HIV-1 consensus B Gag protein (cat# ARP-12425). Peptides are produced as 15-mers with 11 amino acid overlaps. The HCMV pp65 peptide pool is a 138-peptide array spanning the entire pp65 protein of HCMV (ARP-12014). Most of the peptides are 15 amino acids in lengths, with 11 amino acids overlaps between sequential peptides. Both pools of peptides were supplied by the AIDS Reagent Program of NIH (National Institute of Health, USA). We included this information in method section (lines 468-472, page 13)

The sequences of the peptides were not provided in the data sheet of NIH, however these HIV-1 and CMV peptide pools are frequently used by HIV researchers, including us (Nat Med. 2010 Apr;16(4):452-9. doi: 10.1038/nm.2106)

 4. Figures 1 (panels B, C, and D), and 2 (panels A, B, and C): the HIV+ group is redundant and can be deleted (the authors can keep only one set). Answer:  The HIV+ group seems redundant, but we tried to demonstrate the reasoning behind our assessment more clearly. First, we looked at the percentages of cytokine-producing CD4+ T cells (Figure 1, B, C, D) and CD8+ T cells (Figure 2, A, B, C) in response to stimulation (HIV-1, CMV, SEB) in all illicit drug users, regardless of the type of drug used, compared to the PWH group (non-drug users). Subsequently, we looked at the percentage of cytokine-producing cells in the groups of PWH drug users, separately, according to specific drug use in relation to the PWH group (non-drug users). The representation of PWH group in both part of the graph may let the comparison clearer for the readers.  In order to to make it clearer, what we want to show in each figure, we have changed the numbering of figures Figure 1 represents the gate strategy used to evaluate cytokine-producing cells by flow cytometry; Figure 2 represents the percentages of cytokine-producing CD4+ T cells and Figure 3 represents the percentages cytokine-producing CD8+ T cells, after HIV Gag peptide pool, CMV pp65 peptide pool  and SEB 5. Figure 1 panel C IL17/IFN: there seems to be some significant differences (SEB in cocaine + cannabis) and the same applies for Fig.2 panel B (CMV-IL17) and C double positive. I suggest including all p values or a supplemental table for all p values! Answer: Thank you for your comment. We have put all the P values in a supplemental table according to your suggestion so that there is no doubt about the differences in the comparisons found or not between the groups.  The table is attached. If the review 2 think it is important to include this table as supplmental material, we agree. 6. Lines 284-288: “In cannabis and cocaine user….” This part is vague and must be rephrased to convey the right message. Answer: Your comment was very important. We noticed in the paragraph that the information was repeated, which caused doubt when reading the passage. We have removed the repeated statement as you can see on pages 10-11, lines 363-369349. 7. Line 324: the authors should indicate the strengths of their study not just the limitations.Answer: Thank you for your comments, as they have helped us to improve our manuscript. We have included the strengths of our work in the discussion, as follow: Our study has several strengths, including the stratification of PWH who are drug users, categorized as users of cannabis, cocaine, or a combination of cannabis and cocaine, using the urine drug screening test. This stratification made it possible to evaluate the effects of either a single drug or a combination of drugs on antigen-specific T-cell responses. Another important point is that all PWH were on ART with undetectable viral loads, which allowed to avoid the effects of HIV replication. The fact that all PWH were positive for CMV led to the investigation of the effects of drug use on pathogen other than HIV, enriching our work. This is included in lines 404-411, page 11. We also have included the importance of the work on understanding the action of illicit drugs in modulate the specific antiviral immune response from type Th1 to type Th17, in  PWLH under antiviral treatment, as you can see on page 11 and 12, lines 415-418.Minor comments: 1. “In vitro” must be italicized (at line 55 and elsewhere throughout the ms and figure legends). – In vitro was italicized throughout the text of the manuscript.  2. Line 79: Tc17 should be spelled out and abbreviated then used as abbreviated. -  First mention of Tc17 was spelled out on page 2, line 763. Line 102: All individuals drug users → All individual drug users – Thank you for your observation. We did the correction. 4. Significance decimals could be shortened to three digits instead of four throughout the ms. Number of p-value digits was shortened throughout the text. 5. Line 242: PWH individuals → PWH. Correction was done (line 323, page 10).6. Line 277: explain these discrepancies results → explain these discrepancies. Correction was done (line 359, page 10).

Reviewer 3 Report

Comments and Suggestions for Authors

1.    The sample size includes 37 people with HIV undergoing ART (10 non-drug users as control group, and 27 drug-users: 10 cannabis, 7 cocaine and 10 cannabis plus cocaine users). With this small number of people studied, the results obtained may not be representative and clinically relevant.

2.    There is a lack of clinical information on the included subjects, such as, for example, the type of antiretroviral treatment followed, amount of alcohol consumed, amount of cocaine and/or cannabis administered, presence of comorbidities such as diabetes, and chronic immune-mediated diseases.

3.    In the penultimate paragraph of the Discussion it is commented that "we can speculate that the components present in the smoke produced by those drugs could contribute to the immunodeviation to IL-17 producing cells." In the discussion of the results that must be objective, it does not seem appropriate to include "speculations",

Author Response

Reviewer 3 #

 We would like to thank you for your considerations that enabled us to better describe our results throughout the manuscript.

  1. The sample size includes 37 people with HIV undergoing ART (10 non-drug users as control group, and 27 drug-users: 10 cannabis, 7 cocaine and 10 cannabis plus cocaine users). With this small number of people studied, the results obtained may not be representative and clinically relevant.

Answer: We can understand your statement about the sample size, however, we have to consider that this population is difficult to access. In addition, our study group is an important population, as illicit drug users tend to abandon treatment. In this case, the individuals in our study were undergoing treatment and were being followed up by an infectious disease doctor with no viral load.  Therefore, our results represent, within the parameters evaluated in this study, how drug use can act on the development of the specific cellular immune response in PWLH. As we mentioned to reviewer 1, the differences observed in this study, based on statiscal significance, showed that the use of cannabis plus cocaine by PWH increased the percentages of HIV-specific IL-17 producing CD4+ and CD8+ T cells compared to PWH who were non-drug users. Additionally, cocaine use by PWH increased IL-17 CMV-specific CD8+T cell response compared to PWH non-drug users. The elevation in the frequencies of IL-17 cells in PWH drug-user groups could potentially contribute to inflammatory conditions associated with IL-17 overproduction, possibly leading to disease progression. Therefore, these findings are clinically and biologically significant, warranting further investigation.

  1. There is a lack of clinical information on the included subjects, such as, for example, the type of antiretroviral treatment followed, amount of alcohol consumed, amount of cocaine and/or cannabis administered, presence of comorbidities such as diabetes, and chronic immune-mediated diseases.

Answer: Your observation is important. The Table 1 shows the type of antiretroviral medication used by each group (% of each regimen inside the group), CD4 counts and HIV viral load (indetectable levels). To clarify your doubt about alcohol consumed, we did not perform any tests to confirm or exclude the alcohol use in any of the groups of the study.  The use of alcohol was self-declared, such as, the use of illicit drug. However, the use of cannabis and cocaine were confirmed by multidrug test in the urine of all PWH of the study. Thus, the amount of cocaine and/ or cannabis use was not possible to quantify. We would like to inform you, as we did for reviewer #1, that the study subjects were recruited at the time of their medical follow-up. None of them were attending for the first time. All individuals were already taking antiretroviral drugs. The inclusion criteria were declaration of use of illicit drug to form the PWLH drug user group and non-use illicit drug to form the PWLH non drug users group. Also, other criteria to be included in the study was that the individual must be in use of antiretroviral treatment with undetectable viral load and positive serology for CMV.

We use the multi-drug test (ABONTM Multidrug Test – cat no 1155990204) to detect drugs in urine. It is a screening test for the qualitative detection of multiple drugs and drug metabolites in human urine. Patients with positive screening only for cannabis and/or cocaine were included in the respective groups of drug users. Non-use of drugs was also confirmed using the same kit. Thus, patients with negative screening for all drugs and metabolites detected in the toxicological test were included in the control group (PWH). We did not have information about the comorbidities such as diabetes from the majority of PWH in our study. Moreover, at the time of sample collection, none of the PWH should have undergoing an active process of another disease to ensure it does not interfere with the evaluation of the immune response.

  1. In the penultimate paragraph of the Discussion it is commented that "we can speculate that the components present in the smoke produced by those drugs could contribute to the immunodeviation to IL-17 producing cells." In the discussion of the results that must be objective, it does not seem appropriate to include "speculations",

Answer: Thank you for your comment. In the text of the manuscript, we have replaced the word "speculation" with "imply" as you can see on page 11, line 400.

Round 2

Reviewer 2 Report

Comments and Suggestions for Authors

Please see attached report

Comments on the Quality of English Language

minor edits required

Author Response

Answers to reviewer#2

We would like to thank you for your considerations that enabled us to better describe our results throughout the manuscript.

All changes are written in red in the new version of the manuscript.

The revised article by de Castro et al., entitled “ Polarization of HIV- and CMV-specific IL-17 producing T cells in HIV-infected Cannabis plus Cocaine and Cocaine Users” was much improved but still some minor editorial issues exist. These include:

  1. The new title is not clear, and I suggest changing it into: “Polarization of HIV-1- and CMV-specific IL-17 producing T cells among HIV subjects under antiretroviral therapy with and without Cannabis and/or Cocaine usage”

Answer: Thank you for your comment. There is a consensus that when talking about HIV, certain words and language may have a negative meaning for people at high risk for HIV or those who are living with HIV. There is a recommendation of Centre of Disease Control and Prevention (CDC) to use People living with HIV or People with HIV  instead of HIV-infected individuals or HIV subjects. (https://www.cdc.gov/stophivtogether/library/stop-hiv-stigma/fact-sheets/cdc-lsht-stigma-factsheet-language-guide.pdf). International organizations such WHO, UNAIDS, are using People living with HIV (PLWH).

According to the reviewer suggestion and the consideration made about PWH, and that fact that the IL-17 polarization was observed only in the PWH drug users, the title was changed to :

Polarization of HIV-1- and CMV-specific IL-17 producing T cells among People with HIV under antiretroviral therapy with Cannabis and/or Cocaine usage

  1. Abstract: The number of subjects in the four patient groups should be indicated. This was raised in the first revision cycle.

Answer: Sorry for have not made this changed before. We included the number of PWH for each group, lines 27 and 28, page 1.

  1. Figures 4 and 5: A pie chart does not reflect the variability of cytokine production frequencies among subjects within and in between groups. It seems to present the frequency in one individual. The legend should state “mean frequencies” instead of just “frequencies”. For me, it is better to present these frequencies as bar charts with SEM or at least as figures 2 and 3.

Answer: As suggested, we changed the pie chart for individual values representing frequency of cytokine producing cells of each subject of each group and Mean ± Standard Error of the Mean (SEM). The statistical analysis was performed using one-way ANOVA with multiple comparisons and Tukey’s post-hoc testing. All the information was added in the  legends of Figure 4 (pages 8-9, lines 318-325) and  Figure 5 (page 10, lines 361-368) and statistical section (page 14, lines 562-565).

Lines 318-325:

Figure 4. Frequencies of cytokine-producing CD4+ T cells in response to in vitro stimulation with HIV Gag peptide pool (A), HCMVpp65 peptide pool (B), or SEB (C) in each group are represented. Each graph depicts the individual frequency and the mean frequencies ± Standard Error of the Mean (SEM) of single and double cytokine-producing CD4+ T cells from PWH drug users (cannabis = 10, cocaine n = 7, and cannabis plus cocaine n = 10) and non-drug users (n = 10) in response to HIV Gag peptide pool, CMV, and SEB peptides based on IFN-γ, IL-17, or IFN-γ+IL-17 production are represented. For statistical analysis, one-way ANOVA with multiple comparisons and Tukey’s post-hoc testing was used.  p values < 0.05 were considered statistically significant.

Lines 361-368:

Figure 5. Frequencies of cytokine-producing CD8+ T cells in response to in vitro stimulation with HIV Gag peptide pool, HCMVpp65 peptide pool, or SEB in each group are represented. Each graph depicts the individual frequency and the mean frequencies ± Standard Error of the Mean (SEM) of single and double cytokine-producing CD8+ T cells from PWH drug users (cannabis = 10, cocaine n = 7, and cannabis plus cocaine n = 10) and non-drug users (n = 10) in response to HIV Gag peptide pool, CMV, and SEB peptides based on IFN-γ, IL-17, or IFN-γ+IL-17 production are represented.  For statistical analysis, one-way ANOVA with multiple comparisons and Tukey’s post-hoc testing was used.  p values < 0.05 were considered statistically significant.

Lines 562-565:

The results were presented as individual frequency of each cytokine and the mean frequencies ± Standard Error of the Mean (SEM) were shown. For statistical analysis of frequencies, the one-way ANOVA with multiple comparisons and Tukey’s post-hoc testing was used.

In the results section, we included the description of the data  of the new version of figure 4 and figure  5 (page 8, lines 274-282 and page 9, lines 326-338). In the discussion, it was added the text of this results (page 11, lines 444-452).

Lines 274-282:

The CD4+ T cells from the group of PWH users of cocaine showed higher frequency of IL-17 producing cells in response to HIV peptide pool stimulation, compared to frequencies of IFN-g cells and IFN-g+IL-17+ cells (p=0.0012 and p= 0.0002, respectively) (Fig. 4A). No differences were found in the frequencies of cytokine producing CD4+ T cells in response to CMV peptide pool stimulation in all groups (Fig. 4B).  When cells were stimulated with SEB, higher frequencies of IFN-g producing cells were observed in PWH non-drug users (p<0.0001), cannabis users (p=0.0017), and cannabis plus cocaine users (p=0.0141), compared to IL-17 or IFN-g+IL-17 producing cells (Fig.4C).

Lines 444-452:

The frequency analysis of cytokine producing cells in response to HIV peptide pool stimulation, in each group of PWH drug users showed higher levels of IL-17 producing CD4+ T cells in cocaine group and IL-17 producing CD8+ T cells in cocaine plus cannabis group, in comparison to other cytokine producing cells. In relation to CMV specific response, frequencies of IL-17 producing CD8+ cells were higher in cannabis, and cocaine groups compared to frequencies of IFN-γ and IFN-γ+IL-17 producing cells. However, polyclonal stimulation in PWH non-drug user and drug users the frequency of IFN-γ producing cells was predominant, suggesting that the IL-17 response is related to antigen-specific T cell response in PWH drug users.

  1. The reference indicated in response to comment #3 of the original version (Nat Med. 2010 Apr;16(4):452-9. doi: 10.1038/nm.2106) should be added to the revised version.

Answer: We included the reference – See page 16, lines 706-708

  1. In response to comment # 5 of the original revision, I agree to add the p values as a supplemental table.

Answer: The Supplemental Table S1 was included, and it was mentioned at legend of Figure 2 (line 220, page 6) and legend of Figure 3 (line 266, page 7)

  1. Line 357: “stimuli” → “stimulus”,
  • line 368: “to impair” → “to impairment of”,
  • line 417: “Thelper1” → “Th1” and “Thelper17” → “Th17”
  • Line 447: “and or cocaine,” → “and/or cocaine”

Answer: All the corrections was done.

Line 357: “stimuli” → “stimulus”,  new version at line 408.

  • line 368: “to impair” → “to impairment of”, new version at line 419.
  • line 417: “Thelper1” → “Th1” and “Thelper17” → “Th17” , new version at line 473
  • Line 447: “and or cocaine,” → “and/or cocaine”, new version at line 504